# Nighttime O($^1$D) distributions in the mesopause region derived from SABER data

Mikhail Yu. Kulikov[1,2], and Mikhail V. Belikovich[1]

[1]Institute of Applied Physics of the Russian Academy of Sciences, 46 Ulyanov Str., 603950 Nizhny Novgorod, Russia

[2]Lobachevsky State University of Nizhni Novgorod, 23 Gagarin Avenue, 603950 Nizhny Novgorod, Russia

*Correspondence to*: Mikhail Yu. Kulikov (mikhail_kulikov@mail.ru)

**Abstract.** In this study, the new source of O($^1$D) in the mesopause region due to the process OH($v{\geq}5$) + O($^3$P) $\rightarrow$ OH($0{\leq}v^{'}{\leq}v$-5) + O($^1$D) is applied to SABER data to estimate the nighttime O($^1$D) distributions for the years 2003-2005. It is found that O($^1$D) evolutions in these years are very similar to each other. Depending on the month, monthly averaged O($^1$D) distributions demonstrate from 2 to 4 maxima with values up to 340 cm$^{-3}$ which are localized in height (at ~92-96 km) and latitude (at ~20-40$^\circ$S,N and ~60-80$^\circ$S,N). Annually averaged distributions in 2003-2005 have a one weak maximum at ~93 km and ~65$^\circ$S with values of 150-160 cm$^{-3}$ and 3 pronounced maxima (with values up to 230 cm$^{-3}$) at ~95 km and ~35$^\circ$S, at ~94 km and ~40$^\circ$N, at ~93 km and ~65-75$^\circ$N correspondingly. In general, there is slightly more O($^1$D) in the northern hemisphere than in the southern hemisphere. The obtained results are a useful data set for subsequent estimation of nighttime O($^1$D) influence on the chemistry of the mesopause region.

## 1 Introduction

Daytime O($^1$D) is considered to be one of the important chemical minor species of the stratosphere, mesosphere and thermosphere, as it plays a significant role in the chemistry, and the radiative and thermal balance of this region (Brasseur & Solomon, 2005). First of all, formed by photolysis of O$_2$ and O$_3$, O($^1$D) is a mediator involved in the transformation of absorbed solar radiation energy into the heating of this region and, in particular, excitation of N$_2$($v$) and CO$_2$($v$) (Harris & Adams, 1983; Panka et al., 2017). Also, O($^1$D) atoms participate in the reactions of destruction of long-lived greenhouse gases (Baasandorj et al., 2012), CH$_4$ oxidation, and HO$_x$ and NO$_x$ production, for example:

O($^1$D)+N$_2$O $\rightarrow$2NO

O($^1$D)+H$_2$O $\rightarrow$ 2OH

O($^1$D)+H$_2$ $\rightarrow$ H+OH

O($^1$D)+CH$_4$ $\rightarrow$ CH$_3$+OH

O($^1$D)+CH$_4$ $\rightarrow$ H$_2$+CH$_2$O

Moreover, the red line emission from O($^1$D) atoms is one of the most important airglow phenomenon which are used as a diagnostic of the ionosphere, for example, to monitor the electron density and neutral winds in the F region (Shepherd et al., 2019). Therefore, many papers and experimental campaigns are devoted to measurements of features of O$_3$ photolysis to O($^1$D) (Taniguchi et al., 2003; Hofzumahaus et al., 2004).

Until recently, it was believed that the above mentioned processes stop at night as a continuous source of O($^1$D) is absent while the life time of the component is extremely low (less than 1 s). In principle, O($^1$D) can be generated in sprite halos but for a short duration of 1 ms (Hiraki et al., 2004). Recently, Sharma et al. (2015) and Kalogerakis et al. (2016) basing on laboratory experiments proposed that O($^1$D) could be produced in the mesopause region via the process OH($v{\geq}5$) + O($^3$P) $\rightarrow$ OH($0{\leq}v^{'}{\leq}v$-5) + O($^1$D), that is multiquantum quenching of high excited states of OH by collisions with atomic oxygen in ground state.

Last year, Kalogerakis (2019) showed that a new model of O$_2$ A-band, that takes this process into account, describes well (qualitatively and quantitatively) the results of early nighttime rocket measurements of volume emission rate profiles of this

airglow. Thus, he proved that the process OH($v{\geq}5$) + O($^3$P) → OH($0{\leq}v^{'}{\leq}v$-5) + O($^1$D) really took place in nighttime
mesopause, and the produced O($^1$D) distributions can be evaluated from available data.
In this study, the new source of O($^1$D) in the mesopause region is applied to SABER data to estimate the O($^1$D) nighttime
distributions for the years 2003-2005.

## 2 O($^1$D) derivation from SABER Data

All processes used for O($^1$D) determination are summarized in Table 1. Here, we apply the new OH($v$) model of Fytterer et
al. (2019). Their "best-fit model" includes all commonly used production and loss processes of OH($v$) (see Table 1), but
some parameters of the model, in particular, branching ratios of quenching OH($v$)+O$_2$ and rate coefficients of OH($v{\geq}5$) +
O($^3$P) → OH($0{\leq}v^{'}{\leq}v$-5) + O($^1$D) were adjusted with the use of volume emission rate profiles in four different wavelengths
measured by SABER and SCIAMACHY.
Due to low values of chemical lifetimes (less than 1 s), O($^1$D) can be considered in chemical equilibrium:
$$O(^1D) = \frac{k_9 \cdot OH(9) \cdot O(^3P) + k_{10} \cdot OH(8) \cdot O(^3P) + k_{11} \cdot OH(7) \cdot O(^3P) + k_{12} \cdot OH(6) \cdot O(^3P) + k_{13} \cdot OH(5) \cdot O(^3P)}{k_{14} + k_{15} \cdot O_2 + k_{16} \cdot N_2} \qquad (1)$$
Thus, to calculate local value of O($^1$D) we should specify the local concentrations of OH($v$=5-9) and O($^3$P). The mentioned
model lets us to derive the OH($v$) concentrations as the functions of the OH($v$) source due to the reaction (1) ($P_{OH} = k_1 \cdot H \cdot$
$O_3$), air concentration (M), temperature (T), and O($^3$P) concentration:
$$OH(v) = F_v(P_{OH}, M, T, O(^3P)) \qquad (2)$$
To determine O($^3$P) and $P_{OH}$, we use the known (e.g., Mlynczak et al., 2013, 2018) approach for O($^3$P) derivation from the
simultaneous SABER measurements of volume emission rate of (9-7) and (8-6) OH transitions ($VER_{2\,\mu m}$), O$_3$ (9.6 μm), and
temperature (T). The approach employs the chemical equilibrium condition for nighttime ozone. As the result, it is done with
the use of the following system of equations:
$$\begin{cases} P_{OH} = k_2 \cdot O(^3P) \cdot O_2 \cdot M - k_3 \cdot O(^3P) \cdot O_3 \\ VER_{2\,\mu m} = k_4(9,7) \cdot F_9(P_{OH}, M, T, O(^3P)) + k_4(8,6) \cdot F_8(P_{OH}, M, T, O(^3P)) \end{cases} \qquad (3)$$
Thus, we derive the local values of O($^3$P), $P_{OH}$, and OH($v$=5-9) from SABER data with the use of eqs. (2-3) and apply sets of
data (T, M, OH($v$=5-9), and O($^3$P)) to retrieve the local concentrations of O($^1$D) with the use of eq. (1).
The systematic uncertainty of retrieved data is defined by uncertainties in $VER_{2\,\mu m}$, O$_3$, T measurements, and in the rates of
chemical and physical processes included in the OH($v$) model. We reproduced the analysis presented in Fytterer et al. (2019)
(see Sect. 3.4) and took into account the uncertainties of measured data and rate constants which are shown in Table 2. The
third column of the Table demonstrates the uncertainties' individual impact at derived O($^1$D) local concentration. It can be
noted that the most critical for O($^1$D) are the uncertainties in T, rates of reactions (2-3), Einstein coefficients for the $v$ =8-9
states, and $VER_{2\,\mu m}$. The total systematic O($^1$D) uncertainty was obtained by calculating the root-sum-square of all
individual uncertainties. It was found to vary in the range of (37-52)% depending on the pressure level. Due to averaging, the
random error of data presented below is negligible.

## 3 O($^1$D) nighttime distributions

We use the version 2.0 of the SABER data product (Level2A) for the simultaneously measured $VER_{2\,\mu m}$, O$_3$, and T profiles
within the 0.01–0.0001 hPa pressure ($p$) interval (approximately 80–105 km in 2003-2005. We take only nighttime data
when the solar zenith angle χ > 95°. The range of latitudes covered by the satellite trajectory in a month was divided into 20
bins ~ (5.5-8)$^{°}$ each. 1500-3000 single profiles of O($^1$D) concentration fall into one bin during a month of SABER
observations. For each bin we calculate monthly averaged zonal mean < $O(^1D)$ > distributions (hereafter, the angle brackets
are used to denote timely and spatially averaged values). For annually averaged distributions, we use 40 bins ~ 4$^{°}$ each.

Monthly averaged $< O(^1D) >$ distributions in corresponding month of 2003-2005 are shown in Figs. 1–3. Let's analyze the presented data using the distributions in 2003 as an example. Depending on the range of latitudes covered by the satellite trajectory in specified month, the figures show from 2 to 4 maxima which are localized in height (at ~92-96 km) and latitude (at ~20-40°S,N and ~60-80°S,N). The values of the maxima can reach up to 300 cm$^{-3}$ and more in both hemispheres and different months, for example, in January-March and in May-August. Nevertheless, annual cycle of southern $O(^1D)$ demonstrates certain differences from northern one, i.e. many features of $< O(^1D) >$ in the southern hemisphere are not repeated in the northern hemisphere with a shift of 6 months. In particular, the distributions in January-February show 2 pronounced maxima with close values (up to 300 cm$^{-3}$): the first one is at ~95 km and ~50-60°S, the second one is at ~93 km and ~60-80°N. Half a year later (in July-August), we can see 1-2 weak maxima in the southern hemisphere and a strongly pronounced maximum at ~95 km and ~40-50°N. A similar pattern can be noticed comparing the $< O(^1D) >$) distributions in June and December. The satellite trajectory in March and September allows us to observe simultaneously 4 maxima. Note that the southern high-latitudinal maximum (up to 340 cm$^{-3}$) in March does not correspond to the relatively weak northern high- latitudinal maximum in September.

The $< O(^1D) >$ evolutions in 2004-2005 are very similar to 2003. Nevertheless, one can see some differences. First of all, in January-February 2004, there is a pronounced particularity above 60°N below 90 km which does not appear in 2003 and 2005. Kulikov et al. (2019) found similar features in the latitude dependence of nighttime ozone chemical equilibrium boundary (the lower boundary of the altitudinal-latitudinal region where this equilibrium is satisfied (Belikovich et al., 2018; Kulikov et al., 2018)) in January–March 2004 above 60°N and connected it with abnormal dynamics of the stratospheric polar vortex during 2003–2004 Arctic winter. There are additional features also which take place in a specific year, but absent in other two years. In particular, the northern high-latitudinal maximum in January-February 2003 is remarkably higher (by the value) than the ones in January-February 2004-2005. The southern high-latitudinal maximum (up to 340 cm$^{-3}$) in March 2003 corresponds to the same maximum in March 2005 but both maxima are remarkably higher than the one in March 2004. The reverse (relative to December 2003 and 2005) ratio can be observed for the values of southern and northern maxima in December 2004.

Annually averaged $< O(^1D) >$ distributions in 2003-2005 are shown in Fig. 4. There can be seen one weak maximum at ~93 km and ~65°S with values of 150-160 cm$^{-3}$ and 3 pronounced maxima (with values up to 230 cm$^{-3}$) at ~95 km and ~35°S, at ~94 km and ~40°N, at ~93 km and ~65-75°N. In general, there is slightly more $O(^1D)$ in the northern hemisphere than in the southern hemisphere.

**4 Discussion and Conclusion**

According to various early papers (Nicolet, 1959; Ghosh & Gupta, 1970; Shimazaki & Laird, 1970; Harris & Adams, 1983), daytime $O(^1D)$ concentrations at 90-100 km varied in the range of $(10^2$-$10^3)$ cm$^{-3}$. Brasseur & Solomon (2005) published the table (see Table A.6.2.c) where daytime $O(^1D)$ changed from 70 cm$^{-3}$ at 90 km to 140 cm$^{-3}$ at 100 km. The presented results show that monthly and annually mean nighttime $O(^1D)$ concentrations at these altitudes can reach 300 cm$^{-3}$ and 200 cm$^{-3}$, respectively. Thus, nighttime concentrations of $O(^1D)$ are comparable with daytime concentrations of this component and, in principle, can impact noticeably the chemistry and thermal balance of the mesopause region. The analysis of this impact should be carried out with the use of a global 3D chemical transport model of the mesosphere – lower thermosphere. Additionally, it may indicate measurable characteristics of this region that could indirectly confirm the results obtained in this article. In principle, direct evidences of $O(^1D)$ layer existence in nighttime mesopause can be established by *in situ* measurements of $O(^1D)$ airglow at 630 nm which can be carried out, for example, as a part of future WADIS rocket sounding mission (Strelnikov et al., 2019; Grygalashvyly et al., 2019). More detailed analysis is out of this short article scopes.

**Data availability.** The SABER data used in this study can be downloaded from ftp://saber.gats-
inc.com/Version2_0/Level2A/. The presented data can be downloaded from
http://www.iapras.ru/english/structure/dep_240/dep_240.html.
**Author contributions.** Both authors contributed equally to this paper.
**Competing interests.** The authors declare that they have no conflict of interest.
**Acknowledgments.** The work was carried out at the expense of the state assignment #0729-2020-0037. The authors are
grateful to the SABER team for data availability.

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

**Table 1. List of processes.**


| | Process | Rate | Reference |
|---|---|---|---|
| 1 | $H + O_3 \rightarrow O_2 + OH(v)$ | $k_1$ <br> $k_1(v) = k_1 \cdot f(v)$ | Burkholder et al. (2015) <br> Alder-Golden (1997, Table 1) |
| 2 | $O(^3P) + O_2 + M \rightarrow O_3 + M$ | $k_2$ | Burkholder et al. (2015) |
| 3 | $O(^3P) + O_3 \rightarrow 2O_2$ | $k_3$ | Burkholder et al. (2015) |
| 4 | $OH(v) \rightarrow OH(v^{'}) + hv$ | $k_4(v, v')$ | Xu et al. (2012, Table A1) |
| 5 | $OH(v) + N_2 \rightarrow OH(v^{'}) + N_2$ | $k_5(v, v')$ | Adler-Golden (1997, Table 1), <br> Kalogerakis et al. (2011) |
| 6 | $OH(v) + O_2 \rightarrow OH(v^{'}) + O_2$ | $k_6(v, v')$ | Adler-Golden (1997, Table 3), <br> corrected and adjusted by Fytterer <br> et al. (2019) |
| 7 | $OH(v) + O(^3P) \rightarrow H + O_2$ | $k_7(v)$ | Varandas (2004, Table 3, M I) |
| 8 | $OH(v) + O(^3P) \rightarrow OH(v^{'}) + O$ | $k_8(v, v')$ | Caridade et al. (2013, Table 1) |
| 9 | $OH(9) + O(^3P) \rightarrow OH(3,4) + O(^1D))$ | $k_9$ | Fytterer et al. (2019) |
| 10 | $OH(8) + O(^3P) \rightarrow OH(3) + O(^1D))$ | $k_{10}$ | Fytterer et al. (2019) |
| 11 | $OH(7) + O(^3P) \rightarrow OH(\leq 2) + O(^1D))$ | $k_{11}$ | Fytterer et al. (2019) |
| 12 | $OH(6) + O(^3P) \rightarrow OH(\leq 1) + O(^1D))$ | $k_{12}$ | Fytterer et al. (2019) |
| 13 | $OH(5) + O(^3P) \rightarrow OH + O(^1D))$ | $k_{13}$ | Fytterer et al. (2019) |
| 14 | radiative decay of $O(^1D)$ | $k_{14}$ | Burkholder et al. (2015) |
| 15 | $O(^1D) + O_2 \rightarrow O(^3P) + O_2$ | $k_{15}$ | Burkholder et al. (2015) |
| 16 | $O(^1D) + N_2 \rightarrow O + N_2$ | $k_{16}$ | Burkholder et al. (2015) |


**Table 2. List of systematic uncertainties of measured data and rate constants and corresponding uncertainties in derived $O(^1D)$ local concentration.**

| Measured characteristic <br> or rate | Its uncertainty | $O(^1D)$ uncertainty, % |
|---|---|---|
| $VER_{2\,\mu m}$ | 6% | 11-17.5 |
| $O_3$ | 10% | 0.1-8.2 |
| T | from García-Comas et al. (2008) | 0.1-29.7 |
| $k_2$ | from Burkholder et al. (2015) | 14-30.5 |
| $k_3$ | from Burkholder et al. (2015) | 0.7-19.5 |
| $k_{15}$ | from Burkholder et al. (2015) | 2.6-2.8 |
| $k_{16}$ | from Burkholder et al. (2015) | 8-10 |
| $f(9)$ | 0.03 | 1-6 |
| $f(8)$ | 0.03 | 1.4-8 |
| $k_4(9, v')$ | 30% | 12-23.1 |
| $k_4(8, v')$ | 30% | 11-24.6 |
| $k_4(7, v')$ | 30% | 0.6-1.3 |
| $k_4(6, v')$ | 30% | 0.6-1.3 |
| $k_4(5, v')$ | 30% | 0.3-0.9 |



**Figures**

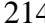

**Figure 1. Monthly averaged O($^1$D) concentration (in cm$^{-3}$) in January-April of 2003-2005.**



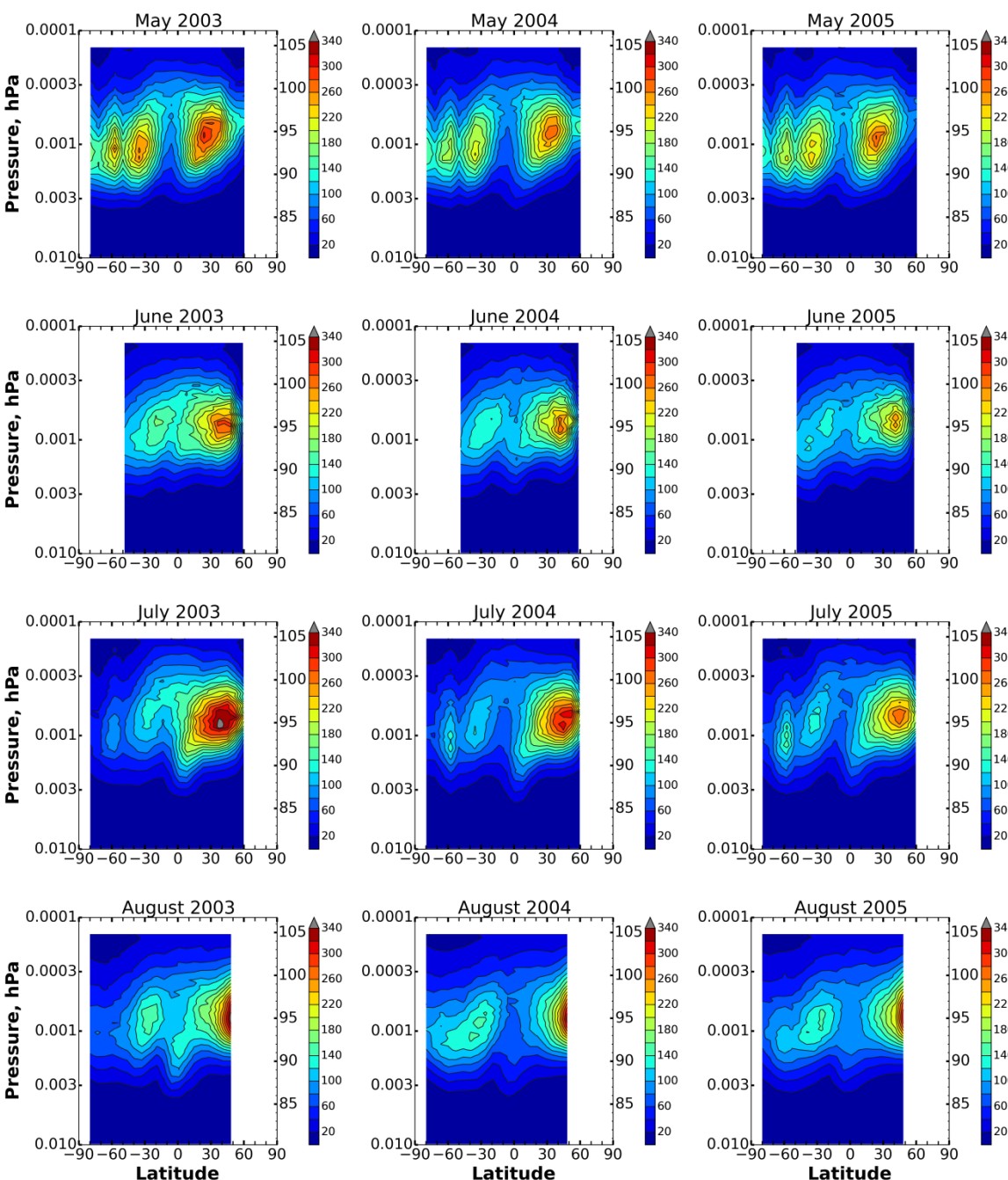

Figure 2. Monthly averaged O($^1$D) concentration (in cm$^{-3}$) in May-August of 2003-2005.




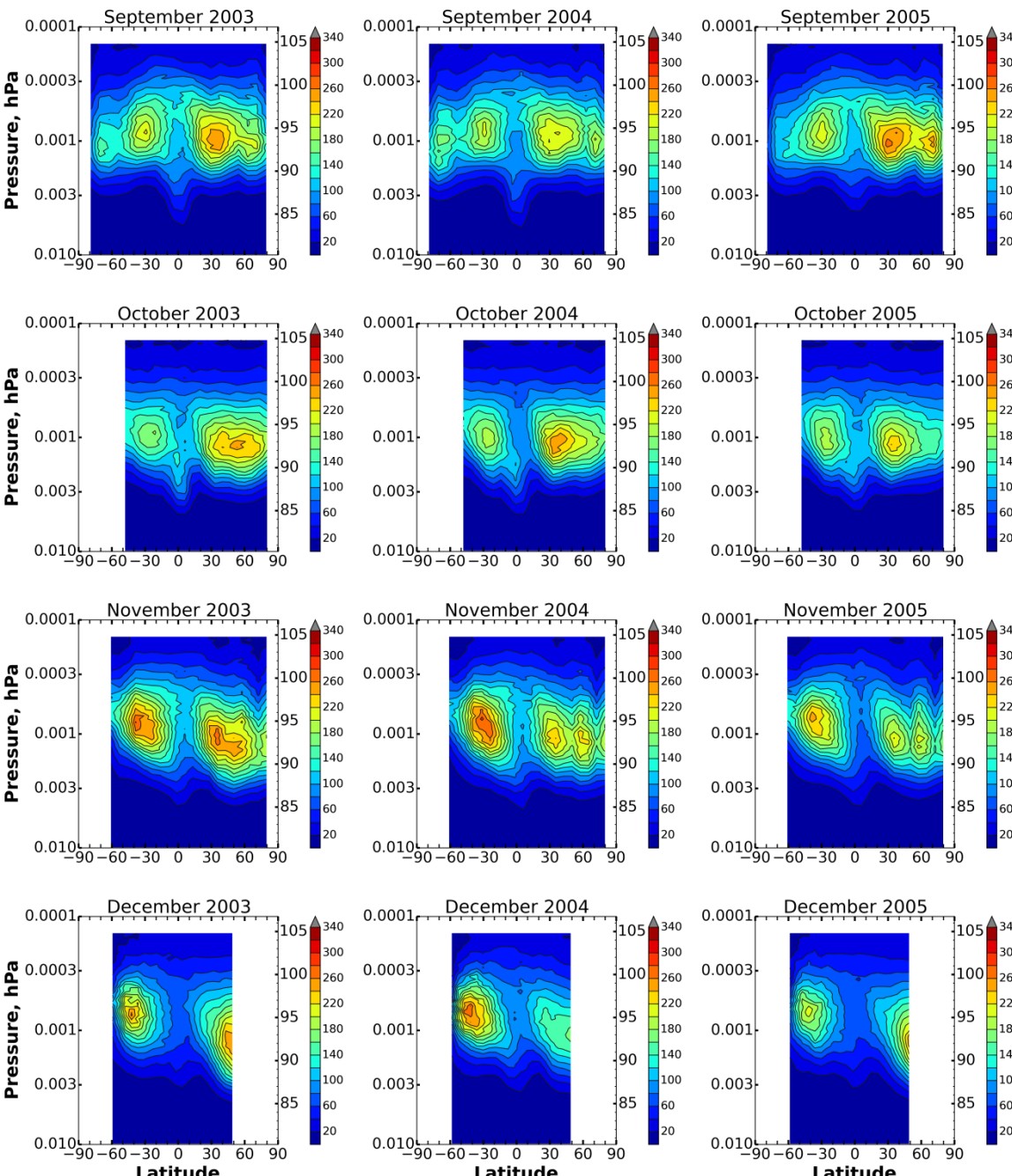

Figure 3. Monthly averaged O($^1$D) concentration (in cm$^{-3}$) in September-December of 2003-2005.



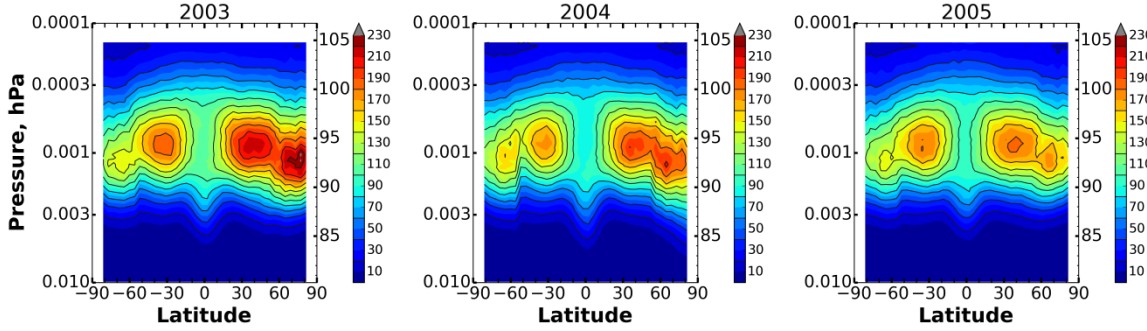


**Figure 4. Annually averaged O($^1$D) concentration (in cm$^{-3}$) in 2003-2005.**