# Peer review of "Nighttime O(1D) distributions in the mesopause region derived from 1 SABER data 2"

_Annales Geophysicae, 2020_

## Short Comment (SC1) · 6 Apr 2020

I consider this paper as HIGHLY SIGNIFICANT! The O(1D) distributions in nighttime mesosphere-mesopause region is characterized by lack of knowledge corrently. For present time this is THE BEST retrieving of given constituent at given region and time. The introduction is informative. The analysis and conclusions of the paper are reasonable. Obtained information is scientifically valuable because it necessary for correct interpretation the Atmospheric band observations and understanding several other chemical processes. The methods and approaches are correct (for present day the retrieving via Fytterer et al. (2019) model is a best way). I found this paper enjoyable to

read (please, correct one minor misprint - numeration of equations should be from (1)).

I recommend it for publication in Annales Geophysicae as it is .

Dr. M. Grygalashvyly

———————————————————

---

## Referee Comment (RC1) · Anonymous Referee #1 · 9 Apr 2020

The manuscript presents a retrieval and a 3-year climatology of nighttime O(1D). The retrieval is based on recent ideas by Kalogerakis, Sharma and co-workers on production of O(1D) through the reaction OH(v>=5) + O(3P). O(1D) is estimated based on primary retrieval products from SABER (essentially OH volume emission rates, O(3P), O3 and T). Central to the analysis is the use of the OH airglow model by Fytterer et al. (2019).

This is indeed a very promising approach with large potential of providing new insights into the chemistry of the mesosphere and lower thermosphere. I recommend the manuscript for publication after an improvement of the discussion of uncertainties

and limitations of the method.

Uncertainties are briefly discussed in the last paragraph of Section 3. Basically, the authors refer to the uncertainty analysis by Fytterer et al (2019) which they have repeated. This needs to be extended. Please summarize the ideas of the uncertainty analysis (based on Fytterer et al.) and state in particular which reaction steps and reaction rates are critical for the uncertainty. This extended paragraph should be moved to the end of Section 2, i.e. the uncertainties of the method should be discussed before the results are presented.

Validation of the O(1D) retrievals is beyond the scope of this manuscript. Nevertheless, I would like the authors to comment on perspectives towards a future validation of these SABER retrievals.

Some minor comments:

Line 9, 11: remove "the" before "values" (3 times)

Line 13: "a useful data set"

Line 14: "on the chemistry"

Line 31: replace "constant" by "a continuous"

Line 34: "via the process"

Line 54: "use the known"

Line 77: "Half a year"

Line 78: "A similar pattern"

Line 83: "There is a pronounced"

Line 84: "found similar features"

Line 84-85: I do not understand the notation "nighttime ozone chemical equilibrium

boundary", in particular the term "boundary". Please clarify.

Line 86: "of the stratospheric"

Line 93: remove "the" before "values" (2 times)

Line 94: remove the word "correspondingly"

Line 98: replace "summarized" by "the total"

Line 108: "use of a global"

---

## Author Comment (AC1) · 21 May 2020

Dear Dr. Grygalashvyly,

Thank you for the supportive comment. We are pleased to see that you value our work as high as we do. It is inspiring to meet someone who is so passionate about subject of our research. The misprint you noted will be duly corrected.

---

## Author Comment (AC2) · 21 May 2020

Below Referee's comments are marked by red, our responses are in black.

Uncertainties are briefly discussed in the last paragraph of Section 3. Basically, the authors refer to the uncertainty analysis by Fytterer et al (2019) which they have re-peated. This needs to be extended. Please summarize the ideas of the uncertainty analysis (based on Fytterer et al.) and state in particular which reaction steps and reaction rates are critical for the uncertainty. This extended paragraph should be moved to the end of Section 2, i.e. the uncertainties of the method should be discussed before the results are presented.

We agree with the comment. In the revised manuscript, the uncertainty analysis was extended. First of all, we added the new Table with the list of systematic uncertainties of measured data and constants and corresponding uncertainties in derived $O(^1D)$ local concentration. The corresponding paragraph was added by a couple of sentences:

«The systematic uncertainty of retrieved data is defined by uncertainties in $VER_{2\,\mu m}$, $O_3$, T measurements, and in the rates of chemical and physical processes included in the OH($v$) model. We reproduced the analysis presented in Fytterer et al. (2019) (see Sect. 3.4) and took into account the uncertainties of measured data and rate constants which are shown in Table 2. The third column of the Table demonstrates the uncertainties' individual impact at derived $O(^1D)$ local concentration. It can be noted that the most critical for $O(^1D)$ are the uncertainties in T, rates of reactions (2-3), Einstein coefficients for the $v$ =8-9 states, and $VER_{2\,\mu m}$. The total systematic $O(^1D)$ uncertainty was obtained by calculating the root-sum-square of all individual uncertainties. It was found to vary in the range of (37-52)% depending on the pressure level. Due to averaging, the random error of data presented below is negligible.»

Furthermore, the paragraph was moved to the end of Section 2.

Validation of the O(1D) retrievals is beyond the scope of this manuscript. Nevertheless, I would like the authors to comment on perspectives towards a future validation of these SABER retrievals.

We agree with the comment. In the revised manuscript, we added a couple of sentences in the end of Sect.4 (Discussion and Conclusion):

«The analysis of this impact should be carried out with the use of a global 3D chemical transport model of the mesosphere – lower thermosphere. Additionally, it may indicate measurable characteristics of this region that could indirectly confirm the results obtained in this article. In principle, direct evidences of $O(^1D)$ layer existence in nighttime mesopause can

be established by *in situ* measurements of O($^1$D) airglow at 630 nm which can be carried out, for example, as a part of future WADIS rocket sounding mission (Strelnikov et al., 2019; Grygalashvyly et al., 2019). More detailed analysis is out of this short article scopes.»

Some minor comments:
Line 9, 11: remove "the" before "values" (3 times)
Line 13: "a useful data set"
Line 14: "on the chemistry"
Line 31: replace "constant" by "a continuous"
Line 34: "via the process"
Line 54: "use the known"
Line 77: "Half a year"
Line 78: "A similar pattern"
Line 83: "There is a pronounced"
Line 84: "found similar features"
Line 84-85: I do not understand the notation "nighttime ozone chemical equilibrium boundary", in particular the term "boundary". Please clarify.
Line 86: "of the stratospheric"
Line 93: remove "the" before "values" (2 times)
Line 94: remove the word "correspondingly
"Line 98: replace "summarized" by "the total"
Line 108: "use of a global"

In the revised manuscript, all these points were corrected accordingly.

---

## Author Response (AR1)

Below Referee's comments are marked by red, our responses are in black.

Uncertainties are briefly discussed in the last paragraph of Section 3. Basically, the authors refer to the uncertainty analysis by Fytterer et al (2019) which they have re-peated. This needs to be extended. Please summarize the ideas of the uncertainty analysis (based on Fytterer et al.) and state in particular which reaction steps and reaction rates are critical for the uncertainty. This extended paragraph should be moved to the end of Section 2, i.e. the uncertainties of the method should be discussed before the results are presented.

We agree with the comment. In the revised manuscript, the uncertainty analysis was extended. First of all, we added the new Table with the list of systematic uncertainties of measured data and constants and corresponding uncertainties in derived $O(^1D)$ local concentration. The corresponding paragraph was moved to the end of Section 2 and was added by a couple of sentences (see lines 63-70 in "Manuscript with marked changes" below):

«The systematic uncertainty of retrieved data is defined by uncertainties in $VER_{2\,\mu m}$, $O_3$, T measurements, and in the rates of chemical and physical processes included in the OH($v$) model. We reproduced the analysis presented in Fytterer et al. (2019) (see Sect. 3.4) and took into account the uncertainties of measured data and rate constants which are shown in Table 2. The third column of the Table demonstrates the uncertainties' individual impact at derived $O(^1D)$ local concentration. It can be noted that the most critical for $O(^1D)$ are the uncertainties in T, rates of reactions (2-3), Einstein coefficients for the $v$ =8-9 states, and $VER_{2\,\mu m}$. The total systematic $O(^1D)$ uncertainty was obtained by calculating the root-sum-square of all individual uncertainties. It was found to vary in the range of (37-52)% depending on the pressure level. Due to averaging, the random error of data presented below is negligible.»

Validation of the O(1D) retrievals is beyond the scope of this manuscript. Nevertheless, I would like the authors to comment on perspectives towards a future validation of these SABER retrievals.

We agree with the comment. In the revised manuscript, we added a couple of sentences in the end of Sect.4 (see lines 117-123 in "Manuscript with marked changes"):

«The analysis of this impact should be carried out with the use of a global 3D chemical transport model of the mesosphere – lower thermosphere. Additionally, it may indicate measurable characteristics of this region that could indirectly confirm the results obtained in this article. In principle, direct evidences of $O(^1D)$ layer existence in nighttime mesopause can be established by *in situ* measurements of O($^1$D) airglow at 630 nm which can be carried out, for example, as a part of future WADIS rocket sounding mission (Strelnikov et al., 2019; Grygalashvyly et al., 2019). More detailed analysis is out of this short article scopes.»

Some minor comments:

In the revised manuscript, all these points were corrected accordingly.

Line 9, 11: remove "the" before "values" (3 times)

Done. See lines 10 and 12 in "Manuscript with marked changes".

Line 13: "a useful data set"

Done. See line 14.

Line 14: "on the chemistry"

Done. See line 14.

Line 31: replace "constant" by "a continuous"

Done. See line 15.

Line 34: "via the process"

Done. See line 35.

Line 54: "use the known"

Done. See line 56.

Line 77: "Half a year"

Done. See line 86.

Line 78: "A similar pattern"

Done. See line 87.

Line 83: "There is a pronounced"

Done. See line 92.

Line 84: "found similar features"

Done. See line 93.

Line 84-85: I do not understand the notation "nighttime ozone chemical equilibrium boundary", in particular the term "boundary". Please clarify.

The sentence was corrected. See lines 94-95.

Line 86: "of the stratospheric"

Done. See line 95.

Line 93: remove "the" before "values" (2 times)

Done. See line 103.

Line 94: remove the word "correspondingly"

Done. See line 104.

Line 98: replace "summarized" by "the total"

Done. See line 68.

Line 108: "use of a global"

Done. See line 118.

[revised manuscript text omitted]